The highly toxic and cryptogenic clinging jellyfish Gonionemus sp. (Hydrozoa, Limnomedusae) on the Swedish west coast

Govindarajan Annette F. 1
http://orcid.org/0000-0003-3116-1029 Källström Björn 2 3 4 bjorn.kallstrom@gmbl.se
http://orcid.org/0000-0002-2579-0841 Selander Erik 2
Östman Carina 5
http://orcid.org/0000-0001-6854-2031 Dahlgren Thomas G. 2 3 6 thda@norceresearch.no
1 Biology Department, Woods Hole Oceanographic Institution , Woods Hole, MA , USA
2 Department of Marine Sciences, University of Gothenburg , Göteborg , Sweden
3 GGBC Gothenburg Global Biodiversity Centre , Göteborg , Sweden
4 Gothenburg Marine Biological Laboratory , Göteborg , Sweden
5 Evolutionary Biology Centre, EBC, Department of Organismal Biology, University of Uppsala , Uppsala , Sweden
6 NORCE Norwegian Research Centre , Bergen , Norway
Reimer James
Electronic publication date: 2019 May 13
Publication date: 2019
Volume: 7
Electronic Location ID: e6883
Received 2019 Jan 18; Accepted 2019 Apr 1
Copyright: © 2019 Govindarajan et al.
Copyright year: 2019
Copyright holder: Govindarajan et al.
License: This is an open access article distributed under the terms of the Creative Commons Attribution License, which permits unrestricted use, distribution, reproduction and adaptation in any medium and for any purpose provided that it is properly attributed. For attribution, the original author(s), title, publication source (PeerJ) and either DOI or URL of the article must be cited.
License URL: https://creativecommons.org/licenses/by/4.0/

Keywords: Sea grass, Zostera, Taxonomy, Biogeography, Climate change, Burn, Nematocyst, Ultrastructure, Microscope, Tentacle

Funding: Swedish Research Council (VR) to Erik Selander “Signals in the Sea” Faculty of Science of Uppsala University Funding for the DNA sequencing analysis Family Foundation and the Borrego Foundation Funding was provided by the Swedish Research Council (VR) to Erik Selander “Signals in the Sea” and from the Faculty of Science of Uppsala University to Carina Östman. Funding for the DNA sequencing analysis was provided by the Kathleen M. and Peter E. Naktenis Family Foundation and the Borrego Foundation. The funders had no role in study design, data collection and analysis, decision to publish, or preparation of the manuscript.

==============================
The clinging jellyfish Gonionemus sp. is a small hydromedusa species known historically from the Swedish west coast but not reported in recent times. This species is thought to be native to the northwest Pacific where it is notorious for causing severe stings in humans and is considered invasive or cryptogenic elsewhere. This year, unlike in the past, severe stings in swimmers making contact with Gonionemus sp. medusae occurred in Swedish waters from a sheltered eelgrass bed in the inner Skagerrak archipelago. To the best of our knowledge, this is only the second sting record of Gonionemus sp. from the Northeast Atlantic—with the first record occurring off the Belgian coast in the 1970s. Stinging Gonionemus sp. medusae have also been recently reported from the northwestern Atlantic coast, where, like on the Swedish coast, stings were not reported in the past. We analyzed sea surface temperature data from the past 30 years and show that 2018 had an exceptionally cold spring followed by an exceptionally hot summer. It is suggested that the 2018 temperature anomalies contributed to the Swedish outbreak. An analysis of mitochondrial COI sequences showed that Swedish medusae belong to the same clade as those from toxic populations in the Sea of Japan and northwest Atlantic. Gonionemus sp. is particularly prone to human-mediated dispersal and we suggest that it is possible that this year’s outbreak is the result of anthropogenic factors either through a climate-driven northward range shift or an introduction via shipping activity. We examined medusa growth rates and details of medusa morphology including nematocysts. Two types of penetrating nematocysts: euryteles and b-mastigophores were observed, suggesting that Gonionemus sp. medusae are able to feed on hard-bodied organisms like copepods and cladocerans. Given the now-regular occurrence and regional spread of Gonionemus sp. in the northwest Atlantic, it seems likely that outbreaks in Sweden will continue. More information on its life cycle, dispersal mechanisms, and ecology are thus desirable.

Introduction

There is increasing concern over the highly toxic cryptogenic clinging jellyfish Gonionemus sp. (Hydrozoa, Limnomedusae) due to outbreaks in scattered temperate coastal areas worldwide, where the jellyfish are either previously unrecorded, or where they have not been observed for decades (Rodriguez et al., 2014; Govindarajan & Carman, 2016; Gaynor et al., 2016; Govindarajan et al., 2017; Marchessaux et al., 2017). These hydromedusae can have a potent sting that causes severe pain and other symptoms to humans (Pigulevsky & Michaleff, 1969; Otsuru et al., 1974; Yakovlev & Vaskovsky, 1993; Govindarajan & Carman, 2016; Marchessaux et al., 2017). As well, they can be lethal to their predators (Carman, Grunden & Govindarajan, 2017).

It appears likely that the current Gonionemus outbreaks are facilitated by anthropogenic transport (Govindarajan & Carman, 2016; Marchessaux et al., 2017). The adult Gonionemus medusae which reach approximately three cm in diameter, have adhesive structures positioned toward the distal ends of their tentacles (Edwards, 1976), which they use to cling to the eelgrass such as Zostera marina (Perkins, 1903; Uchida, 1976). Thus, while the medusa occasionally swim out of the eelgrass meadows, natural or anthropogenic medusa dispersal, while possible, may not be the primary mechanism for its spread. Gonionemus sp. has a complex life history that includes minute benthic asexual stages (Perkins, 1903; Kakinuma, 1971; Uchida, 1976) that may be amenable to human-mediated transport on ship hulls (Tambs-Lyche, 1964), shellfish (Edwards, 1976), and debris (Choong et al., 2018).

An understanding of the dispersal history and spread of clinging jellyfish has been hampered by a complex taxonomic history. The name Gonionemus vertens Agassiz, 1862 has been used recently to refer to “clinging jellyfish” from throughout the northern hemisphere but was originally described from material collected in Puget Sound, the North East Pacific (Agassiz, 1862). In the Atlantic the clinging jellyfish were originally described as G. murbachii Mayer, 1901, and were considered distinct from G. vertens (Mayer, 1901). They were later synonymized (Kramp, 1959) and the Atlantic populations were hypothesized to have been founded by anthropogenic introductions from the Pacific (Tambs-Lyche, 1964; Edwards, 1976; Bakker, 1980), although this was not accepted by all. Based on consistent morphological characters, some authors either maintained the murbachii name (Rottini, 1979) or considered the two forms to be subspecies (Naumov, 1960). Govindarajan et al. (2017) found that differences in mitochondrial COI sequences were also consistent with the vertens—murbachii forms; but noted that these differences do not correspond to the Atlantic—Pacific division suggested by Naumov (1960). Owing to their episodic nature and the lack of continuity in observations of late 19th and early 20th century G. murbachii and contemporary populations in the G. murbachii type locality, Govindarajan et al. (2017) conservatively referred to the more toxic, putative murbachii lineage as Gonionemus sp. until the taxonomy can be further clarified.

Gonionemus sp. has been previously reported from Scandinavian and North Sea waters (reviewed in Bakker, 1980; Wolff, 2005), but we are not aware of any stings associated with past observations. Here, using morphological and molecular evidence, we document blooms of the highly toxic lineage Gonionemus sp. in the summer of 2018 associated with a sheltered eelgrass (Z. marina) bed on the Swedish west coast. We report the first case of a Gonionemus sp. envenomation in Scandinavian waters and discuss the possible origins of these apparently new and highly toxic Gonionemus sp. populations. We also suggest that warmer than average sea surface temperatures may have contributed to the 2018 Gonionemus sp. outbreaks.

Materials and Methods

Sample collection and field observations

The first reports of an unknown stinging medusae came from swimmers through media on 27th July 2018 (SVT, 2018). Several swimmers had been stung at Knuten on the northeastern (leeward) side of the island Tjörn (58.0782°N; 11.7065°E, Fig. 1). On two occasions, the 2nd and 18th of August, we sampled medusae by snorkeling in the eelgrass bed with a small hand-held net (120 × 150 mm, mesh size 0.5 mm; JBL GmbH & CO, Neuhofen, Germany). Medusae were transported live in aerated 20 l tanks to the laboratory for analysis and maintained on a diet of copepods and frozen Artemia naupli larvae, fed once every day. Samples were preserved in 96% ethanol for DNA analysis.

Figure 1 Map of the study area indicating new records of Gonionemus sp.

Red dots are records associated with stings. Historical records of Gonionemus sp., which are not associated with stings, indicated by blue dots. The blow up shows the area on the Swedish coast where stinging Gonionemus were found during 2018. Near the location where they were found are two international harbors, shown as black filled triangles. The sea surface temperature monitoring station in Åstol is indicated by a star.

Medusa size and nematocyst identifications

The diameter of the Gonionemus medusae was measured by imaging the uncontracted medusae when resting on the bottom of a white plastic box with 50 mm of natural sea water using a DSLR camera (Nikon D7100; Nikon Corporation, Tokyo, Japan). A plastic millimeter ruler in the box was used for reference to measure the bell diameter in Image J (Schneider, Rasband & Eliceiri, 2012). Some of the Gonionemus medusae and their nematocysts were examined and photographed with a Leica M205C (Leica Microsystems, Wetzlar, Germany) stereomicroscope and a Leitz DMRBE (Leica Microsystems, Wetzlar, Germany) light microscope (LM) equipped with interference-contrast optics, 100×/1.30 PL, fluotar objectives. Both microscopes were connected to the digital photo equipment Leica application suite, version 3.8 (LAS V3.8). Measurements on different medusa structures were made in the stereomicroscope, and measurements on the nematocysts were made from the LM using live tissue carefully squashed under a cover glass. All pictures and measurements are from living hydromedusae.

The nematocysts were identified by size, structure and shape of their undischarged and discharged capsule and shaft, and on the spine-pattern of the shaft. The classification system and nematocyst nomenclature of Östman (2000, and references therein) was used.

Temperature records

Sea surface temperature data were downloaded from the Swedish repository for environmental monitoring data (Swedish Meteorological and Hydrological Institute (SMHI), 2018). The closest monitoring station with sufficient resolution and duration was “Åstol,” 23 km from the collection site (57.922°N; 11.590°E, Fig. 1). Sea surface temperature from 1986 to 2018 was binned into monthly averages. The monthly mean temperatures for 2018 were graphically superimposed to identify anomalies.

Phylogeographic analysis

Molecular procedures and analyses were conducted at the Woods Hole Oceanographic Institution (Woods Hole, MA, USA) except where indicated. Genomic DNA was extracted from 15 preserved hydromedusae collected from the leeward side of Tjörn Island, Skåpesund (Fig. 1) using a DNeasy Blood & Tissue Kit (Qiagen, Los Angeles, CA, USA) according to the manufacturer’s protocol. A ∼650 base pair portion of the mitochondrial COI gene was amplified and sequenced using primers from Folmer et al. (1994) using the approach described in Govindarajan et al. (2017). PCR conditions were 3 min at 95 °C; 35 cycles of 95 °C 30 s; 48 °C 30 s, 72 °C 1 min; and 5 min at 72 °C. PCR products were visualized on a 1% agarose gel stained with GelRed, purified with QIAquick PCR Purification Kit (Qiagen, Los Angeles, CA, USA) according to the manufacturer’s protocol, and quantified using a NanoDrop 2000 spectrophotometer (Thermo Fisher Scientific, Waltham, MA, USA). Purified products were sequenced in both directions (Eurofins, https://www.eurofins.com/). An additional specimen was amplified using a similar protocol in Sweden and sent for sequencing using the GATC LightRun Barcode service (www.eurofinsgenomics.eu). Sequence chromatograms were evaluated and assembled using Geneious version 9.0.5 (https://www.geneious.com/). Assembled sequences were aligned with sequences representing the seven haplotypes in Govindarajan et al. (2017; Table 1). Representatives of additional haplotypes from Gonionemus sp. sequences that were deposited on GenBank after Govindarajan et al. (2017) were identified in a preliminary alignment and were then added to the alignment dataset with the Swedish sequences. Alignments were conducted using Clustal W (Larkin et al., 2007) in the Geneious platform with default parameters. The alignments were confirmed by eye and the ends were trimmed to 501 base pairs to standardize sequence length and facilitate a direct comparison with the analysis conducted by Govindarajan et al. (2017) and the new GenBank sequences that were also that length. Neighbor-joining trees based on Kimura two-parameter distances (to be consistent with previous analyses; Zheng et al., 2014; Govindarajan et al., 2017) were constructed using PAUP* 4 (Swofford, 2003) accessed through Geneious.

Table 1 Gonionemus sp. COI haplotypes.

Haplotype	Genbank accession number for representative sequence	Known localities	References	
Haplotype 1	KF962139	China (unspecified)	He et al., unpublished GenBank entry	
Haplotype 2	KY437853	Pacific coast of Japan; Yellow Sea	Govindarajan et al. (2017)	
Haplotype 3	KY437979	Sea of Japan (Vostok Bay)	Govindarajan et al. (2017)	
Haplotype 4	KY437944	Sea of Japan (Vostok Bay, Amur Bay); Pacific coast of Japan; Northwest Atlantic coast of USA (New Hampshire, Massachusetts, Rhode Island, Connecticut); Sweden	Govindarajan et al. (2017); This study	
Haplotype 5	KY437888	Northwest Atlantic coast of USA (Massachusetts, New Hampshire)	Govindarajan et al. (2017)	
Haplotype 6	KY437842	Northwest Atlantic coast of USA (Massachusetts, Rhode Island, Connecticut)	Govindarajan et al. (2017)	
Haplotype 8	MK158933	Sweden	This study	
Haplotype 9	MK158944	Northwest Atlantic coast of USA; Sweden	This study	
Haplotype 10	MH020743	China (Yellow Sea)	Liu & Dong, unpublished GenBank entry	
Haplotype 11	MH020707	China (Bohai Sea)	Liu & Dong, unpublished GenBank entry	
Haplotype 12	MH020652	China (unspecified)	Liu & Dong, unpublished GenBank entry	
Haplotype 13	MH020717	China (Yellow Sea)	Liu & Dong, unpublished GenBank entry	
Haplotype 14	MH020722	China (Yellow Sea)	Liu & Dong, unpublished GenBank entry	
Haplotype 15	MH020725	China (Yellow Sea)	Liu & Dong, unpublished GenBank entry	
Haplotype 16	MH020640	China (unspecified)	Liu & Dong, unpublished GenBank entry	
Note:

Known haplotypes of Gonionemus sp. COI and locations where they have been documented. The GenBank accession numbers are for the representative sequences used to construct the neighbor—joining tree in Fig. 6.

Results

Sample collection and field observations

Medusae were collected at two occasions from the Skåpesund location (Fig. 1) and identified morphologically as Gonionemus sp. (Fig. 2). Medusae possessed adhesive pads characteristic of the genus Gonionemus located toward the distal ends of their tentacles (Figs. 2 and 3, more detailed morphology in Figs. S1–S6), which allow them to “cling” to the eelgrass. Similar to Gonionemus sp. from the Northwest Atlantic, Northwest Pacific, and the Sea of Japan (Kakinuma, 1971; Govindarajan et al., 2017), medusae were relatively flat and had relatively thin, dull orange—brown gonads.

Figure 2 Gonionemus sp. Macromorphology of medusae and tentacles.

Medusae in apical dorsal (A–C), lateral (D and E), and oral ventral view (F and G), showing gonads, radial canals, ring-canal, bell-rim flaps/lappets, statocysts, manubrium, tentacles with nematocyst batteries and adhesive pads, tentacle base tentacle with tentacle-canal and yellow streak, and velum. Abbreviations: arrows, point at adhesive pads; brf, bell-rim flap/lappet; gmd, developing male gonad; gf, female gonad; m, manubrium; rac, radial canal; ric, ring-canal; s, statocyst; sto, stomach; stoa, stomach attachment; tb, tentacle base; tc, tentacle-canal; v, velum; ys, yellow streak. Photo credits: Carina Östman (A–F), Ulf Jondelius (G).

Figure 3 Gonionemus sp. Micromorphology of tentacles and nematocysts.

(A–C) Tentacle parts with nematocyst batteries and adhesive pads. Note dark pigmented mid-line in tentacles, yellow-reddish pigments in batteries and around adhesive pad. (Inset B) LM. Euryteles, note shaft and tubule. (D–K) LMs. Tentacle nematocysts. Undischarged and discharged microbasic euryteles and small microbasic b-mastigophores. Note shaft, tubule, lid and apical capsule opening (*). (E) Microbasic eurytele with broad, rod-shaped shaft, pointed apically and microbasic b-mastigophore with slightly bent, narrow shaft, following the convex capsule side. (F and G) Microbasic b-mastigophores. Note shaft and tubule pattern. (H–J, inset) Discharged microbasic euryteles. Note broad shaft with spined distal swelling, rounded lid, difference in diameter of shaft and distal tubule. (J) Note spine pattern on distal tubule. (I and K) Microbasic b-mastigophores. Narrow shaft with unclear spine-pattern. Abbreviations: *marks apical capsule opening; ap, adhesive pad; b, b-mast, microbasic b-mastigophore; dt, distal tubule; eu, eurytele; l, lid; nb, nematocyst battery; sh, shaft; sp, spines; tu, tubule. Photo credits: Carina Östman.

One of us (BK) was stung several times while skin diving on the first sampling date. The stings left red marks at the site of contact and produced marked pain for several hours and feelings of unease throughout the first night afterward. As reported elsewhere (Pigulevsky & Michaleff, 1969) stinging sensations where felt throughout the night even at places on the body where no direct contact had occurred. Local newspapers also reported stings in other swimmers, with similar outcomes and in a few cases the victims had strong reactions that demanded medical attention (Aftonbladet, 2018).

Medusa size distributions

Our size distribution data were limited to two points in time, not including the time of medusa release or the time of disappearance. This restrained our ability to assess the growth rate. The data we obtained on the Gonionemus sp. population indicated slight growth over the 16-day period between sampling dates (p < 0.001, Fig. 4). At the initial time point, the mean diameter was 9.8 ± 2.7 mm. A total of 16 days later, at the second time point, the mean size was 11 ± 1.8 mm suggesting an average growth rate of 0.08 mm per day.

Figure 4 Size distribution of Gonionemus sp. on August 2 and August 18.

Size increase by 1.2 mm in the 16 days between sampling, corresponding to a growth rate of 0.08 mm d−1 (p < 0.05). Each histogram contains the bell diameters of 120 individuals.

Detailed medusa and nematocyst morphology

Gonionemus sp. possesses a well-developed transparent velum (inward projecting rim of tissue; Figs. 2A–2D). Four narrow radial canals form a noticeable cross centrally inside the subumbrella cavity (Figs. 2B and 2C). The stomach with connecting manubrium (tube-like projection with the mouth) is centrally attached to the cross-region of the radial canals (Figs. 2B–2D). The gonads are arranged along most of the length of the radial canals (Figs. 2A, 2B and 2D). Mature female gonads are light yellow-brown; each gonad is folded into six to eight broad bulbs (Fig. 2A; Figs. S5A–S5D). In the folds between the gonad-bulbs, pores are present, from which eggs are ejected. The male gonads are darker brownish-red and each folded into 9–13 smaller bulbs (Figs. 2B and 2D; Figs. S6A–S6D). As the gonads mature, more and larger gonad-bulbs are developed.

Figure 5 Temperature data from Åstol, adjacent to the locations where Gonionemus sp. was found in 2018.

The black line shows the monthly mean temperature ± standard deviations (shaded area) from 2000 to 2018. The red line with open circles shows the monthly mean temperature during 2018.

Figure 6 Gonionemus sp. Neighbor-joining tree of COI haplotypes based on Kimura two-parameter distances.

Haplotype numbering for haplotypes 1–7 corresponds to those in Govindarajan et al. (2017). Haplotypes 8–16 are newly presented here based on Swedish specimens and Genbank (Table 1).

Around 45–58 slender tentacles are attached to the subumbrella rim close to the ring canal, which surrounds the bell close to the velum (Figs. 2A–2G). Contracted tentacles are stubby (Fig. 2D) and are less than half the length of extended tentacles (Figs. 2F and 2G). Close to or at a short distance from the tentacle tip, a small bending is present on each tentacle, caused by the presence of an adhesive pad (Figs. 2D, 2F and 2G). The adhesive pad is located to one side of the tentacle and causes the tentacle to bend, thus pointing outward (Figs. 2A, 2F and 2G, detailed view in Fig. 3A–3C). One or two statocysts are present between each tentacle pair (Fig. 2E; Fig. S2F).

The gracile tentacles are in their mid-region black colored along most of their length (Figs. 3A–3C) and are armed with ring-shaped nematocyst batteries. Batteries with closely packed nematocysts form rings around the tentacle (Fig. 3A). Small patches of nematocysts are scattered between the nematocyst rings, most clearly visible on the black pigmented mid-streak of a tentacle (Figs. 3A and 3B). Toward the tentacle bases the nematocyst batteries are less dense, sometimes spiral-formed or missing (Figs. S1B and S2D). A yellow pigmented streak is prominent at each tentacle base seen in dorsal view (Fig. 2E; Fig. S2A). The yellow streak is less obvious on the tentacle bases seen in oral view (Figs. S2D and S2E).

Two nematocyst types, microbasic euryteles, and microbasic b-mastigophores, are present in the nematocyst batteries around the tentacles (Figs. 3D–3K). The euryteles are larger and by far the more abundant. Some small microbasic b-mastigophores were loosely scattered among the euryteles. Euryteles are also densely present at the tentacle bases and close to the ring-canal (Figs. S2B and S3). At the manubrial undulating rim euryteles were abundant but loosely scattered on the remaining manubrium (Figs. S4E and S4F). The capsules of both euryteles and the b-mastigophores are broad, rounded basally and slightly narrower apically (Fig. 3E). The inverted eurytele shaft is broad, rod-shaped with pointed apical tip. The pattern of the shaft is caused by its long, inverted spines, all pointing toward the apical capsule opening with its lid. The inverted tubule makes slightly oblique coils to the long capsule axis and almost fills the whole capsule, except for its basal end. The narrow shaft of the small microbasic b-mastigophore is slightly bent, following the convex capsule side (Figs. 3E–3G). Discharged eurytele shaft is broad, rod-shaped with distal swelling armed with long spines (Figs. 3H and 3I, inset). The prominent rounded lid at apical capsule, and the difference of the diameter of shaft and distal tubule are obvious (Figs. 3I and 3J). On discharged microbasic b-mastigophores no clear spine-pattern on the narrow shaft and no obvious difference between the diameter of distal tubule and shaft tubules are visible (Fig. 3I and 3K).

Additional morphological details are presented in Figs. S1–S6.

Temperature

The 2018 spring and summer temperatures in Åstol were anomalous relative to the previous 28 years (Fig. 5). The spring temperatures were approximately 2 °C cooler than during 1986–2018; while the summer temperatures were approximately 2 °C warmer than 1986–2018.

Phylogeographic analysis

DNA Sequences were obtained for 16 Swedish Gonionemus sp. medusae and submitted to GenBank (accession numbers MK158929–MK158944). These 16 sequences comprised three haplotypes. Nine medusae possessed one haplotype, six medusae possessed a second haplotype, and a single medusa possessed a third haplotype. The Swedish sequences were aligned with representatives of each of the Gonionemus sp. haplotypes described in Govindarajan et al. (2017) and additional haplotypes found in subsequently available GenBank sequences. These newer GenBank sequences included one representative from New Jersey on the USA mid-Atlantic coast (accession number KY451454; Gaynor et al., 2016) and 104 sequences from three Chinese locations (accession numbers MH020640–MH020743; Liu & Dong, unpublished GenBank entry).

An initial alignment and neighbor-joining tree of the new Chinese sequences from GenBank showed that they comprised nine haplotypes (Fig. 6). Haplotypes were labeled by number following Govindarajan et al. (2017) and new haplotypes were given new numbers. Of the nine haplotypes, one matched Haplotype 9, one matched Haplotype 4, and seven were unique for Sweden. One sequence representing each of the seven unique haplotypes were selected for the analysis with the Swedish unique sequences. The single New Jersey sequence matched one of the Swedish haplotypes, as described below.

A neighbor-joining tree of the Swedish sequences and the unique COI haplotypes was generated (Fig. 6). We found that the most abundant Swedish haplotype (found in nine out of 16 specimens) exactly matched Haplotype 4 from Govindarajan et al. (2017) that was possessed by medusae from the Northwest Atlantic (from the states of Connecticut, Rhode Island, Massachusetts, and New Hampshire along the northeastern USA coast) and the Northwest Pacific (including the highly toxic Vladivostok-area populations from the Sea of Japan). The second Swedish haplotype, termed “Haplotype 9” here and found in six out of 16 specimens, matched the haplotype from New Jersey.

Discussion

We documented a bloom of the highly toxic clinging jellyfish Gonionemus sp. associated with severe stings to humans on the Swedish west coast. To our knowledge, this is the first record of clinging jellyfish envenomations to humans from this region. The symptoms reported by one of the authors (BK) are consistent with those described from the Northwest Atlantic and Sea of Japan (Mikulich & Naumov, 1974; Michaleff, 1974; Yatskov, 1974; Govindarajan & Carman, 2016).

Clinging jellyfish have been previously reported from European Atlantic, North Sea, and Mediterranean coasts, as well as the northwestern Atlantic, the northwestern Pacific and Sea of Japan (reviewed in Govindarajan & Carman, 2016; Govindarajan et al., 2017; Marchessaux et al., 2017). The records of clinging jellyfish in Europe are sporadic. In Atlantic coastal waters, observations date back to the early 1900s (Bakker, 1980), and in the Mediterranean possibly back to the 1870s (as Cosmotira salinarium; Duplessis, 1879). Gonionemus sp. has also been reported from several aquaria with Atlantic and Mediterranean source water (reviewed in Edwards (1976) and Bakker (1980)). However, in contrast to western Pacific and Sea of Japan populations, where there is a long record of severe stings, stings to humans have not been reported to our knowledge from European populations until 2016 (from the French Mediterranean coast; Marchessaux et al., 2017).

The history of clinging jellyfish in European Atlantic waters is comparable to that along the Northwest Atlantic US coast and may similarly indicate a new, cryptic invasion of a more toxic form. Both regions have a history of episodic clinging jellyfish sightings, but no record of stings until recently. However, the existence of multiple species and highly episodic life cycle make drawing conclusions difficult. Our genetic analysis confirmed our morphological identification, placing the Swedish form into the Gonionemus sp. clade that includes the highly toxic phenotype. Our morphological observations are consistent with historical European observations of the apparently less toxic form (G. murbachii), but it seems likely that toxicity varies within the Gonionemus sp. clade (Govindarajan et al., 2017) so this discrepancy does not rule out a new introduction.

Additional sampling and analysis of nuclear markers will be required to fully solve the Gonionemus “zoogeographic puzzle.” However, our COI data provide several new insights. One of the three Swedish haplotypes (Haplotype 4 in Fig. 6) is also found in the northern Northwest Atlantic and the western Pacific/Sea of Japan regions which contain highly toxic individuals and may indicate a common origin. Another of our haplotypes (Haplotype 9 in Fig. 6) has also been reported from New Jersey, USA, which is in the mid—Northwest Atlantic region. Gonionemus sp. was first reported in New Jersey in 2016 (Gaynor et al., 2016). Our analysis could indicate an independent origin of the New Jersey population relative to the northern Northwest Atlantic populations. Our third haplotype (Haplotype 8 in Fig. 6), found in only one individual, was unique.

Interestingly, our haplotype tree also shows that the Pacific region contains the greatest number of haplotypes (10) but only one of these (Haplotype 4) is found outside of the region. This result is consistent with a scenario where a subset of the ostensibly native Pacific diversity inoculated other regions. However, the observation of several haplotypes in Sweden and elsewhere that have not yet been found in the Pacific, in combination with the historical record of sightings from these same regions, suggests that we cannot rule out that the Northwestern Atlantic and Mediterranean regions contain native diversity, either instead of or in addition to introduced lineages.

There are ample pathways and opportunities for Gonionemus sp. to be introduced to the Swedish coast. The life cycle of Gonionemus sp. includes minute polyp and cyst stages (Perkins, 1903; Kakinuma, 1971; Uchida, 1976) that could have easily arrived unnoticed. In a genetic survey of epifauna, Gonionemus sp. was recently identified from the North American Pacific coast on tsunami debris originating from Japan (Choong et al., 2018). This suggests that polyp, frustule, and or cyst stages are capable of long-distance transport on anthropogenic surfaces. There are two larger international harbors near our study site, Wallhamn and Stenungsund (Fig. 1); thus, it is quite possible that a highly toxic lineage arrived attached to ship hulls. Furthermore, there are many records of Gonionemus sp. occurring in public aquaria, where they presumably establish from polyp stages accompanying materials brought to the aquaria (Tambs-Lyche, 1964).

Another factor that may have played a role in the 2018 Swedish clinging jellyfish outbreak is temperature. Water temperature is a critical factor initiating seasonal hydrozoan polyp activity (Calder, 1990). The year of the outbreak (2018) was exceptional in that it had both an approximately two degrees colder than average spring and a two-degrees warmer than average summer. Either or both of these anomalies could have facilitated the Gonionemus sp. outbreak. Gonionemus sp. medusae are produced by polyps, which may arise from frustules or cysts (Perkins, 1903; Uchida, 1976). In a detailed study of the life cycle and development of G. vertens, Kakinuma (1971) observed that when polyps were kept at 20 °C with access to food they released medusae that developed to 10–12 mm in diameter in 5–6 weeks. Both temperature and salinity have also been implicated in affecting Russian Sea of Japan populations (Yakovlev & Vaskovsky, 1993). Surface water temperature data available to us (Fig. 5) originated from a temperature monitoring station located in a less sheltered area and in deeper water than the area with Zostera-beds where the current outbreak occurred (Fig. 1). We therefore hypothesize that in the area of the observed outbreak the surface water temperature was above 20 °C for most of July and August, which would have allowed the release and development of medusae from polyps present in the area.

It seems probable that, similar to the US Atlantic coast, Gonionemus sp. will spread to new sites along the North Sea coast and potentially pose hazards to both humans and ecosystems. Regular monitoring and surveys will be crucial for providing warnings to protect bathers and others from potentially harmful interactions. Our two data points for size distribution indicated a ∼12% increase in size over 16 days (Fig. 4). Temperature and food availability are likely factors affecting growth rate and if comparable with Kakinuma (1971), the size distribution in the studied population would imply that the medusae release started roughly around 5 weeks earlier, or in late June.

More information is also needed to understand the impact of Gonionemus sp. in eelgrass communities, but the morphological features we observed provide some insight into their ecological roles. Gonionemus sp. medusae spend much of their time “clinging” to eelgrass with their adhesive pads. Free (unattached) tentacles are often extended into the ambient current. This pattern of passive drifting of tentacles is typically seen in hydromedusae that are ambush predators (Madin, 1988; Colin, Costello & Klos, 2003). Tethering to the seagrass will also increase encounter rate with prey if there is a current passing the tethered medusa. The nematocyst types we observed, euryteles and microbasic b-mastigophores, and their arrangement in raised clusters on the tentacles suggest that Gonionemus sp. feeds on hard-body prey such as crustaceans (Purcell & Mills, 1988). This is consistent with reports that Gonionemus sp. medusae feed on small zooplankton such as copepods (Mills, 1983 (for G. vertens)) and observations in laboratory cultures that they feed on copepods and Artemia nauplii (A. Govindarajan, C. Östman, 2018, personal observation).

Intriguingly, Gonionemus sp. may mediate the interactions of other species and cause mortality in non-prey organisms. For example, along the Northwest Atlantic coast in Massachusetts, Carman, Grunden & Govindarajan (2017) found that Gonionemus sp. was consumed by a native spider crab but not by the invasive green crab. The authors also found that Gonionemus ingestion resulted in crab death when large numbers of jellyfish were consumed. Thus, Gonionemus sp. may potentially impact native ecosystems via differential predation by a native species (spider crabs) that may lead to a decline of that species, while avoidance of Gonionemus by a destructive invasive species could potentially facilitate its dominance.

Conclusions

We documented the presence of the cryptogenic limnomedusa Gonionemus sp. from an eelgrass bed at the Swedish west coast during the summer of 2018. The presence of these medusae were linked to several severe stings in local bathers. Using mitochondrial COI sequences, we showed that the Swedish medusae belong to the same clade as highly toxic populations previously found in the Sea of Japan and the northwestern Atlantic. We also reported detailed features of the medusa morphology using light microscopy, including details of the nematocysts. We suggested that the outbreak at the Swedish west coast is linked to the exceptionally warm summer of 2018 following either a climate-driven range shift or a direct introduction to the area via shipping activity. Given the harmful stings associated with the medusae and the high risk of additional colonization along the Swedish coast, further investigations on this species are warranted.

Supplemental Information

Supplemental Information 1 Gonionemus sp. Additional morphological data.

Each supplementary figure (1–6) show different aspects of medusae morphology.

Click here for additional data file.

Hans Hällman is gratefully acknowledged for sampling assistance and Ulf Jondelius kindly provided the photograph for Fig. 2G.

Additional Information and Declarations

Competing Interests

Author Contributions

Data Availability

The authors declare that they have no competing interests.

Annette F. Govindarajan conceived and designed the experiments, performed the experiments, analyzed the data, contributed reagents/materials/analysis tools, prepared figures and/or tables, authored or reviewed drafts of the paper, approved the final draft.

Björn Källström conceived and designed the experiments, performed the experiments, analyzed the data, contributed reagents/materials/analysis tools, prepared figures and/or tables, authored or reviewed drafts of the paper, approved the final draft, collected samples in the field.

Erik Selander conceived and designed the experiments, analyzed the data, contributed reagents/materials/analysis tools, prepared figures and/or tables, authored or reviewed drafts of the paper, approved the final draft.

Carina Östman analyzed the data, contributed reagents/materials/analysis tools, prepared figures and/or tables, authored or reviewed drafts of the paper, approved the final draft, conducted detailed ultrastructure work.

Thomas G. Dahlgren conceived and designed the experiments, analyzed the data, contributed reagents/materials/analysis tools, prepared figures and/or tables, authored or reviewed drafts of the paper, approved the final draft.

The following information was supplied regarding data availability:

Data is available at Genbank (accession numbers MK158929–MK158944).

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
