# Peer review of "The highly toxic and cryptogenic clinging jellyfish Gonionemus sp. (Hydrozoa, Limnomedusae) on the Swedish west coast"

_PeerJ, doi:10.7717/peerj.6883_

## Round 0.1 · original submission · Minor Revisions

I have heard back from two reviewers, both of whom were very positive about your submission. Both reviewers have also provided constructive comments, none of them major, but which will, in my opinion, help make you manuscript better. Please consider their comments.

As well, reviewer 1 notes that you can contact them should you need some references they mentioned in their review.

I look forward to seeing a revised version.

·

Basic reporting

Some parts are not enough citations.

Experimental design

Sampling data is not enough for estimate growth rate of Gonionemus sp.. If possible, the authors should add more data during all months when medusae appear.


And please add more detail of culturing methods (sizes of tanks, feeding frequency).

Validity of the findings

The authors should discuss why Gonionemus sp. appeared in Swedish waters. Water temperature is one of important trigger for medusa budding. See Kakinuma (1971).

Additional comments

The authors examined medusa growth rate and identified using morphology and molecular method with beautiful photos. I think the paper will be acceptable after minor revision but please consider growth rate and condition of medusa budding.

Reviewer 2 ·

Basic reporting

This is an important contribution to the taxonomy, biogeography and the knowledge of sting-associated records of hydromedusae in general, and the genus Gonionemus in particular. The text is well written, concise and unambiguous, and the English is clear. The question is clear and well contextualized, especially considering the previous paper by Govindarajan et al. (2017), which have discussed the possible origins and scenarios of invasions of lineages within Gonionemus. The authors present detailed descriptions with relevant information on morphology and cnidome, which is accompanied by high quality, exceptionally clear figures. Methods are well-described and sufficient to address the question proposed. Raw data on COI haplotype sequences was supplied.

Experimental design

My only concern here is related to the absence of Table 1, which was not supplied with the submission. I am not sure if it was my mistake (I couldn’t find it), but a Table is cited in line 140. I believe it is very important that the authors include a Table describing the provenance of the material used in the analysis, identifying the ones used for morphological description, and the ones from which COI sequences were obtain. Additionally, I believe it is very important that the authors identify the haplotypes in this table, to make it easier for the reader to understand Figure 6 and the discussion that is related to it. Without a reference of a table with the description of the material, their locality and haplotype designation, it is very difficult to follow the discussion provided by the authors. I also suggest that the authors include the same information about the sequences from GenBank and from Govindarajan et al. (2017).

Validity of the findings

No comment

Additional comments

Another suggestion I have is related to the identification of the species Gonionemus sp. The authors provide such a detailed morphological description, with sequences from several localities, that I was wondering if it wouldn’t be possible to improve their contribution to the taxonomy and systematics of this group. Is it still not possible to refer to this species as G. murbachii? Or at least provide a more solid delimitation for this lineage? In my opinion, a discussion concerning the morphological diagnostic characters of the species and their relationship with other closely related species would be relevant.

Additional minor comments:

- Please correct the year of the paper Govindarajan et al. in the references section.
- Table 1 was cited in line 140 but was not provided with the submission.
- Please remove the additional tentacle in line 3 of the legend of Figure 3.
- Please include a comma after ric and before ring canal in the legend of Figure 3.
- In the legend of figure 3 the radial canal is referred as rac but in the figure it is referred as ra.
- Line 149: I suggest the authors replace the word the for in
- Lines 180-181: the authors describe the male gonad indicating Figures 3B and 3D, but in these figures, the gonads are identified as developing gonads (gd), and not as male gonads. I believe this needs more clarity.
- Legend of Supplementary Figure 6: remove the word in before A,B in line 4.
- Please exclude the word with in line 262.
- Line 291: this sentence is not entirely clear and needs modification.

---

## Round 0.2 · Minor Revisions

I have gone over the paper myself, and find you have addressed all the scientific points well. I have noted some small English mistakes, and attach them here in a MSWord file. Please check these edits - I imagine this will be easy, and I will soon be able to accept this work. I look forward to seeing your resubmission.

---

## Round 0.3 · accepted · Accept

Thank you for the final English edits; your work is now ready to be moved into production. Congratulations!

#